# Effect of Supplementation of Lambs with Whole Cottonseed: Impact on Serum Biomarkers and Infection by Gastrointestinal Parasites under Field Conditions

**DOI:** 10.3390/metabo13030398

**Published:** 2023-03-08

**Authors:** Vitoldo Antonio Kozlowski Neto, Elizabeth Moreira dos Santos Schmidt, Camila Peres Rubio, Naiara Mirelly Marinho da Silva, Renata Tardivo, Ciniro Costa, Paulo Roberto de Lima Meirelles, José Joaquín Cerón, Asta Tvarijonaviciute, Alessandro Francisco Talamini do Amarante

**Affiliations:** 1School of Veterinary Medicine and Animal Science, São Paulo State University (FMVZ, UNESP), Rua Prof. Doutor Walter Mauricio Correa s/n, Unesp Campus Botucatu, Botucatu 18618-681, SP, Brazil; 2Interdisciplinary Laboratory of Clinical Analysis, Interlab-UMU, University of Murcia, Regional Campus of International Excellence “Campus Mare Nostrum”, Espinardo, 30100 Murcia, Spain; 3Department of Animal Nutrition and Breeding, School of Veterinary Medicine and Animal Science, São Paulo State University (FMVZ, UNESP), Rua Prof. Doutor Walter Mauricio Correa s/n, Unesp Campus Botucatu, Botucatu 18618-681, SP, Brazil; 4Institute of Biosciences, São Paulo State University (UNESP), Rua Prof. Dr. Antônio Celso Wagner Zanin, 250, Distrito de Rubião Junior, Botucatu 18618-689, SP, Brazil

**Keywords:** *Eimeria* spp., gastrointestinal nematodes, antioxidant, oxidant, biochemical analytes, integrated crop-livestock system, sheep

## Abstract

The purpose of this trial was to evaluate serum levels of oxidative stress biomarkers and biochemical analytes in crossbred lambs during the rearing phase in an integrated crop-livestock system (ICLS) to control gastrointestinal parasites. The experiment used 36 crossbred lambs (cross: Ile de France × White Dorper × Texel) divided into two groups. The WCS group was supplemented with whole cottonseed (WCS), and controls had no supplementation. Body weight, blood collection, and fecal analysis of nematode eggs and *Eimeria* oocysts counting per gram of feces were performed for each animal within 84 days of experiment. The following serum analytes were determined: total protein, albumin, globulin, cholesterol, haptoglobin, and 10 oxidative stress biomarkers: cupric reducing antioxidant capacity, ferric reducing ability of plasma, trolox equivalent antioxidant capacity, thiol, uric acid, paraoxonase-1, total oxidant status, ferric-xylenol orange, advanced oxidation protein products, and reactive oxygen metabolites derived compounds. The inclusion of WCS suggested the benefit in controlling infection as well as inducing an increase in antioxidants and a decrease in oxidants in lambs naturally infected by gastrointestinal parasites. The combination of WCS and ICLS could be a useful tool in controlling gastrointestinal parasite infection without affecting the production performance.

## 1. Introduction

Sheep farming faces a constant battle against gastrointestinal nematodes (GIN) and *Eimeria* spp. infections, which leads to considerable economic losses. *Haemonchus contortus* followed by *Trichostrongylus colubriformis* are the main GIN infecting sheep raised in Brazil [1,2] and associated with *Eimeria* spp.; they can have clinical and subclinical disease [3]. Lambs are more susceptible and can be infected soon after birth, thus resulting in the elimination of many oocysts in feces within a few weeks of life [4]. Thus, alternatives for the control of gastrointestinal parasites are needed.

The production of sheep in an integrated crop-livestock system (ICLS) enables strategies that increase the income of those who produce milk, meat, and grains; it decreases the seasonality of animals for slaughter [5]. Furthermore, the use of adequate nutrition associated with pastures in an ICLS aids in the control of GIN infection [6]. Animal supplements can improve the productive performance in an ICLS. Whole cottonseed (WCS) is a potential component in diet formulation. WCS is a byproduct of the cotton ginning industry (*Gossypium* L.). It combines high energy with high proportion of protein and effective fiber for ruminant supplementation [7,8]. It has been described as a satisfactory feed ingredient in growing sheep [9] and can modulate the immune system [10].

The imbalance between oxidants and antioxidants either in cells or in the body are defined as oxidative stress [11,12]. Oxidative stress is seen in pathology as well as normal homeostasis; it affects cellular macromolecules especially proteins, lipids, and deoxyribonucleic acids [11]. When there is an uncontrolled production of oxidants such as those from reactive species derived from oxygen, nitrogen, and/or others the organism offers an efficient mechanism to control and restore balance through antioxidants (endogenous and exogenous) that protect, repair, intercept, and prevent oxidative stress damage [12].

In animal health, biomarkers are less widespread and less developed than in human health. However, when applied from the One Health perspective, animal biomonitoring could provide important information, thus supporting the prevention and management of health risks [13]. Antioxidant, oxidant, trace elements, and acute phase proteins biomarkers should be used together, thus becoming important tools when assessing, monitoring, and identifying the oxidative stress associated with diseases [12]. Nevertheless, to the best of our knowledge, there remains a lack of data in the literature regarding the use of the panel of oxidative stress biomarkers targeting the health of sheep when infected by gastrointestinal parasites in combination with the inclusion of WCS in the diet in an ICLS. We hypothesize here that the supplementation of lambs with WCS could be useful to control gastrointestinal (GI) parasites in an ICLS. This could lead to changes in serum analytes specially those related to redox status. Thus, the purpose of this trial was to evaluate changes in GI parasites, body weight, weight gain, and serum levels of redox biomarkers status analytes during the rearing phase in crossbred lambs naturally infected by GIN and *Eimeria* spp. with and without dietary inclusion of WCS in an ICLS.

## 2. Experimental Design

### 2.1. Experimental Area Description

This experiment was conducted between December 2018 and July 2019 at Lageado Experimental Farm (22°51′01″ S and 48°25′28″ W; altitude, 777 m) belonging to FMVZ-UNESP, Botucatu, São Paulo, Brazil. According to the Köppen climate classification system, the region’s climate is Cfa type, which covers 6.5% of the Brazilian territory and 33.4% of the São Paulo state territory. The Cfa type is a warm temperate (mesothermic) humid climate [14,15]. The average accumulated monthly precipitation is highest (260.7 mm) in January and lowest (38.2 mm) in August. The average monthly temperature ranges from 23.2 °C in February to 17.1 °C in July [16].

The experimental area is part of an ICLS established since 2010. Before the present study, the area was used to produce maize (*Zea mays* L.) or soybean (*Glycine max* (L.) Merr.) silage in summer/autumn and black oat (*Avena strigosa* Schreb) oversown in winter/spring with different treatments made with intercropped marandu palisade grass (*Urochloa* (syn. *Brachiaria*) *brizantha* cv. Marandu), black oat, pigeon pea (*Cajanus cajan* (L.) Millsp.), and aruana guinea grass (*Megathirsus* (syn. *Panicum*) *maximum* cv. Aruana). The black oat pastures were grazed by lambs during the winter [6,17].

Soybean (cv. BMX Potência RR) was sown in December 2018. The aruana guinea grass was intercropped with soybean as recommended [18]. After mechanical harvesting (soybean and aruana guinea grass) in March 2019, the pastures of aruana guinea grass reached its ideal management point (presenting 95% of light interception on average). This was grazed by lambs from 8 May to 31 July 2019.

### 2.2. Animals and Management

The study population comprised 36 crossbred (Ile de France × White Dorper × Texel) female lambs during the rearing phase with a mean age of 105 days, mean body weight of 25.6 ± 5.4 kg acquired from a commercial farm located approximately 160 km from Botucatu, São Paulo, Brazil. This study was approved by the Faculty’s Animal Experimentation Ethics Committee of the São Paulo State University—FMVZ, UNESP, Botucatu (0099/2019—CEUA).

Before starting the experimental trial, the animals were tagged and vaccinated against *Clostridium* spp. (Poli-Star^®^, Vallée, Montes Claros, Brazil). Fecal samples were collected for counting of nematode eggs and *Eimeria* spp. oocysts per gram of feces (EPG and OPG, respectively) [19]. All animals were treated subcutaneously with 1% moxidectin (Cydectin^®^, Zoetis, São Paulo, Brazil) as a single application at 0.2 mg/kg on 6 May 2019. There was oral treatment with 5% toltrazuril suspension (Baycox^®^, Elanco, São Paulo, Brazil) due to an outbreak of eimeriosis on 26 June 2019, as a single application at 20 mg/kg.

Eighteen lambs were housed in four pens (two pens with five animals and two pens with four animals). They were fed concentrate, maize silage, and WCS. Another eighteen lambs were also housed in four pens (two pens with five animals and two pens with four animals). They were fed similarly but without WCS. Details about the diet ingredients are presented in Appendix A. All lambs were released from the pens and allocated in their respective ICLS paddocks daily at 7:00 a.m. At 5:00 p.m., they returned to a covered shed lined with rice straw with eight pens each 5 m × 5 m (25 m^2^). The animals always had free access to water. The two groups were balanced as closely as possible by body weight, and the lambs were adapted to the diets for 14 days prior the sampling. At the paddocks, the grazing of the aruana guinea grass was rotational with a fixed stocking rate. This was distributed in groups as previously determined for pens in the covered shed. The area was divided by an electric fence into 24 paddocks 16 m × 30 m (480 m^2^); 12 paddocks for each group were rotated in the same three paddocks during the trial where they grazed for seven days each.

The diets were formulated using the Small Ruminant Nutrition System (SRNS), computer program for sheep [20]. The estimated weight gain was approximately 200 g/day [21].

### 2.3. Body Weight, Fecal and Blood Samples

Feces and blood samples were obtained from each animal every 14 days. Body weight was recorded at the same time. The samplings were: 0 days (8 May 2019), 14 days (22 May 2019), 28 days (5 June 2019), 42 days (19 June 2019), 56 days (3 July 2019), 70 days (17 July 2019), and 84 days (31 July 2019). The fecal samples, blood samples, and body weights at 0, 14, 28, 42, 56, 70, and 84 days were obtained at 6:00 a.m. with the animals fasting in the shed.

#### Egg and Oocyst Counting

Fecal samples were collected directly from the rectum of each lamb, stored in a labeled plastic bag, and transported in thermal boxes. The fecal counting of EPG and OPG were determined using the modified McMaster technique with a sensitivity of 100 eggs per gram of feces (EPG) [19,22]. The 2 g of feces were macerated with 58 mL of saturated NaCl solution at a density of 1.2. The sample was then filtered with gauze or sieve followed by filling the right and left side of the McMaster chamber. We then counted the two cells subdivided by dashes. Each counted egg or oocyst represented 100 nematode eggs or *Eimeria* spp. oocysts per gram (g) of feces.

### 2.4. Biochemical Profile

Blood samples were collected from the jugular vein in plain tubes with gel separators (BD Vacutainer^®^ Blood Collection Tube; Becton, Dickinson and Company, Holdrege, NE, USA). After centrifugation (1500× *g* for 5 min) sera were stored in Eppendorf microtubes (Eppendorf^®^, Hamburg, Germany) at −80 °C until laboratory analysis of the biochemical analytes and biomarkers of oxidative stress. Serum haptoglobin (Hp) concentrations were measured via a hemoglobin binding assay using a commercially available colorimetric method (Haptoglobin Tridelta^®^ phase range, Tridelta Development Ltd., Maynooth, Country Kildare, Ireland). Total serum protein and albumin were measured using an automated analyzer (Olympus AU600, Olympus Diagnostic Europe GmbH, Ennis, Ireland) following the instructions of the manufacturer using Olympus commercial kits. The estimated concentrations of globulins were calculated by the difference between total proteins and albumin concentrations. Total serum cholesterol was determined using a commercial kit (Beckman Coulter^®^ Inc., Fullerton, CA, USA).

### 2.5. Biomarkers of Oxidative Stress

The total antioxidant capacity (TAC) in serum was determined with three different assays [23]: Trolox equivalent antioxidant capacity (TEAC), ferric reducing ability of plasma (FRAP), and cupric reducing antioxidant capacity (CUPRAC) according to methods described [24,25,26]. The thiol antioxidative effects were determined via an automated method based on Ellman’s method [27]. Individual antioxidants such as uric acid and paraoxonase-1 (PON-1) were also evaluated. The uric acid concentration was determined with a commercially spectrophotometric method (OSR6698 Beckman Coulter AU analyzers, Nyon, Switzerland) based on a protocol described previously [28]. Serum PON-1 was determined using p-nitrophenyl acetate as the substrate in an automated clinical chemistry analyzer (Olympus AU2700, Olympus Diagnostica GmbH, Hamburg, Germany) using an adaptation of a previously described assay [29].

Oxidant biomarkers such as advanced oxidation protein products (AOPP), a marker of oxidative damage to proteins, was based on di-tyrosine containing cross-linked proteins and oxidatively modified albumin [30]. Total oxidant status (TOS) was determined via the oxidation of the ferrous ion–o-dianisidine complex [31]. Ferric-xylenol orange (FOX) was determined via the ferrous oxidation by xylenol orange [32]. Reactive oxygen metabolites derived compounds (d-ROMs) were determined by an assay [33] that monitors the N-diethyl-paraphenyldiamine radical cation concentration. All sera analyses used an automated chemistry analyzer (Olympus AU600, Olympus Diagnostic Europe GmbH, Ennis, Ireland).

### 2.6. Statistical Analysis

All variables were first assessed for normality with graphic analysis of histogram, QQ plot, and a Shapiro-Wilkes test. The data that did not present a normal distribution (EPG and OPG) were transformed to log10 (x + 1). All data were analyzed by analysis of variance via the Statistical Analysis System-SAS^®^ Studio (SAS Institute Inc., Cary, NC, USA) with the General Linear Model (GLM). Tukey’s test at a 5% significance level was used to compare means. The EPG and OPG transformed results were reported in tables with arithmetic means (±standard deviation). The graphs used raw data with medians (interquartile range). The results of the variables that had normal distributions were reported in tables with arithmetic means (±standard deviation); the graphs plotted the arithmetic means (±standard error of the mean). A *p*-value of < 0.05 was considered statistically significant for all statistical analysis.

## 3. Results

### 3.1. Natural Infection by Gastrointestinal Parasites

Gastrointestinal nematodes eggs and *Eimeria* spp. infections were detected in lambs during the trial. Significant days effects for EPG and OPG counting were observed during the trial for both groups (Figure 1). The lambs had higher EPG counts at 0 days. The EPG counting decreased for both groups at 14 days and over the next samplings. Low OPG counts were observed at 0 days for both groups (Figure 1). However, there was a significant increase in OPG counting for the following days with a significant peak at 42 days and a significant decrease due to treatment with toltrazuril between 42 and 56 days. The OPG significantly decreased until 70 days. There was a significant increase for both groups at 84 days. The WCS group had a significantly lower general mean of OPG.

### 3.2. Body Weight and Average Daily Weight Gain

The lambs in the WCS and control groups had a similar body weight gain over the experimental period (Figure 2) without significant differences between the groups. Significant days effects were observed for body weight gain for both groups. At the beginning of the trial the average body weight for the WCS group was 25.82 (±5.65) kg and 25.42 (±5.17) kg for the control group, and at the end it was 36.48 (±6.56) kg for the WCS group and 36.82 (±6.32) kg for the control group. The lambs of the WCS group had significantly lower average daily weight gain (ADG) between 42 and 56 days as well as a significantly lower ADG for the lambs in the control group between 56 and 70 days. There was no diet effect (*p* = 0.391) for ADG, however, there were significant group versus days interactions (*p* = 0.028) and days effects (*p* < 0.0001) (Appendix A).

### 3.3. Biochemical Analytes

Biochemistry analytes for control and WCS lambs are presented in Figure 3. The Hp results in serum are below the detection limit of the assay for all animals at all days (data not shown). The lambs in the control group had significantly higher albumin concentrations throughout the trial (Appendix A). The total protein concentrations had significant oscillations throughout the trial in both groups, but the albumin concentrations were higher in the WCS group at the end of the trial versus the beginning. In addition, the globulin estimated concentrations were not significantly different between both groups during the experimental period. The lambs of both groups had significantly higher cholesterol concentrations at 28 and 84 days.

### 3.4. Antioxidant Biomarkers

Antioxidant serum biomarkers for control and WCS lambs are presented in Figure 4 and Appendix A. The lambs of the WCS group had significantly higher TEAC concentrations than the control group after 28 days. No significant differences were observed for FRAP and CUPRAC concentrations between groups. Thiol significantly increased for both groups at 70 and 84 days. Uric acid concentrations had variations for the lambs of both groups but were not significantly different during the experimental period or between groups. The lambs of both groups had significantly lower PON-1 concentrations at the beginning of the trial (0 days), but the WCS group had significantly higher PON-1 concentrations than the control group from 28 to 56 days.

### 3.5. Oxidant Biomarkers

Serum oxidant biomarkers for control and WCS lambs are presented in Figure 5 and Appendix A. The WCS group had significantly higher means for AOPP concentrations that significantly decreased for both groups at the end of the trial. Total oxidant status (TOS) for the WCS group had significantly decreased values at the end of the trial (84 days). The lambs in the control group had significantly higher FOX concentrations than the WCS group during the trial. The WCS group had significantly higher concentrations for d-ROMs than the control group, although its concentrations significantly decreased for both groups from 28 to 56 days.

## 4. Discussion

At the beginning of the experiment, a higher general mean of EPG counting was observed which reduced along the experimental period. High means of EPG counting were observed at the beginning of an ICLS trial with crossbred male lambs with a subsequent decrease in the EPG counting throughout the experiment [6]. The lambs were naturally co-infected by *Eimeria* spp. with a low number of *Eimeria* spp. oocysts at the beginning of the trial. However, there was a peak of OPG counts at 42 days for both treatments. Lambs raised in high densities contribute to the transmission of this protozoan, thus causing a significant elimination of oocysts in the feces [4]. The treatment protocol performed between 42 and 56 days with toltrazuril was effective in controlling the coccidia infection and decreasing the OPG counts. Besides the treatment with toltrazuril, bedding changes as well as cleaning of pens and troughs helped control the infection and reduced the environmental contamination.

Two factors that could have played major roles in decreasing the EPG and OPG counts were the good quality of the diet and the inclusion of WCS because immunity possibly developed efficiently to control the GI parasites infection, thus again increasing the body weight gain [2,6]. The supplementation with WCS had significant effects for several categories of sheep that were naturally infected by GI parasites [34].

There was a significantly lower overall mean of OPG counts for the WCS treatment in this trial, thus suggesting a positive effect for WCS supplementation to control *Eimeria* spp. infection. This effect could be attributed to the presence of gossypol in the WCS, which is an antinutrient polyphenolic compound secreted by pigment glands distributed throughout the cotton plant with higher concentrations in the seeds [35,36]. In addition, gossypol has been reported to have an inhibitory action on enzymes and on the growth of *Trypanosoma cruzi* [37] as well as an anti-amoebic effect on *Entamoeba histolytica* [38]. Although gossypol could be highly toxic to monogastric animals, ruminants are more resistant to the compound due to ruminal fermentation, binding with gossypol, and the consequent reduction in toxic side effects [8,36].

In our study, a higher albumin concentration was observed for the control group with a significant decrease in albumin and total protein concentrations in both groups at 56 days, thus demonstrating the side effects and the body late recovery reaction to *Eimeria* spp. and GIN co-infection. There could be changes in serum proteins in animals with parasitic infections caused by GIN [2,39,40] and by *Eimeria* spp. in sheep [41,42]. Albumin concentrations increased at the end of the trial (84 days) when compared to 0 days for the WCS group. Besides reducing the OPG and EPG counts, the inclusion of WCS in the diet may have prevented structural GI mucosal changes and the loss of albumin into the GI tract. It could have also increased food digestibility and nutrients absorption [43]. In the present study the inflammatory response could not be evaluated as the haptoglobin concentrations were below the limit of detection for all lambs.

An impairment of the intestinal absorption of lipids could lead to diminished appetite and low digestibility of food, which are negative effects of GI parasite infection [44]. No changes in cholesterol concentrations were found in our study. Prior work [45] concluded that there were no consistent changes in biochemical analytes in sheep supplemented with cottonseed cake, although increases in cholesterol levels were reported in lambs under heat stress supplemented with WCS [46].

Biomarkers of oxidative stress provide data regarding the monitoring of different procedures to reduce the parasite load and GIN in sheep [40,47,48,49], stress monitoring [50], and evaluation of diets fed to lambs [51] as well as the intake of cottonseed [52] in the oxidative status of sheep.

The TEAC, FRAP, and CUPRAC assays involve several antioxidant molecules and were used to evaluate TAC. Although FRAP and CUPRAC concentrations did not change, the higher TEAC concentration in lambs supplemented with WCS could reflect an increase in the antioxidants related to a decrease in GI parasites [50]. Increases in TEAC and CUPRAC concentrations were reported in lambs naturally infected by GIN after 70 days of an anthelmintic treatment and under the influence of the ICLS; there was a negative correlation of these biomarkers with EPG [40]. Cottonseed oil is rich in tocopherol and has an antioxidant activity by reacting against free radicals [36,53]. Lactating dairy cows fed with 15% WCS inclusion had higher plasma α-tocopherol concentrations. Although the increase of α-tocopherol in blood could have antioxidant effects, a limitation of this study was that plasma α-tocopherol levels were not determined.

At the end of the trial TOS and FOX concentrations showed a significant decrease in the WCS group. There are literature reports demonstrating increased oxidants concentrations in sheep infected with GIN [47,48,49], which induce local intestinal oxidative stress associated with an inflammatory infiltrate in the mucosa of the abomasum. These effects could cause systemic changes depending on the intensity of infection and changes in the redox status with the purpose of cellular protection [40]. Thus, the inclusion of WCS in the diet reduced the production of oxidants by decreasing the parasite burden at the end of the trial.

In summary, the sum of the co-infection by GIN and *Eimeria* spp. led to limitations for the lambs including an initial poor performance of the animals. Furthermore, the *Eimeria* spp. infection played a role in this scenario although the lambs reached the expected daily weight gain at the end of the trial. Our current scenario aims for sustainable sheep production including animal welfare, thus decreasing the use of unnecessary anthelmintic treatments that could contribute to the development of parasite resistance while diminishing the environmental impacts caused by animal production. According to the findings of this experimental trial, the WCS inclusion in lambs’ diets could be considered an additional alternative aiming to control GI parasite infections. Further studies should be conducted with different percentages of WCS inclusion in the diet.

## 5. Conclusions

The WCS inclusion in this trial suggested a benefit to controlling the natural infection by GIN as well as inducing a decrease in oxidants such as TOS and FOX and an increase in the antioxidant TEAC in lambs. The combination of WCS and ICLS could be a useful tool in controlling gastrointestinal parasite infection without affecting the production performance.

## Figures and Tables

**Figure 1 metabolites-13-00398-f001:**
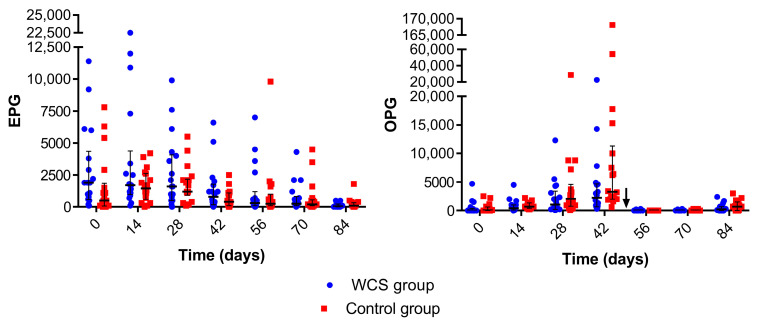
Median (interquartile range) of eggs per gram of feces (EPG) and *Eimeria* spp. oocysts per gram of feces (OPG) of crossbred lambs naturally infected with gastrointestinal parasites and receiving two different diets (whole cottonseed and control) at seven different samplings (0–84 days). There was an interval of 14 days between each sample collection. The arrow indicates the treatment with 5% toltrazuril.

**Figure 2 metabolites-13-00398-f002:**
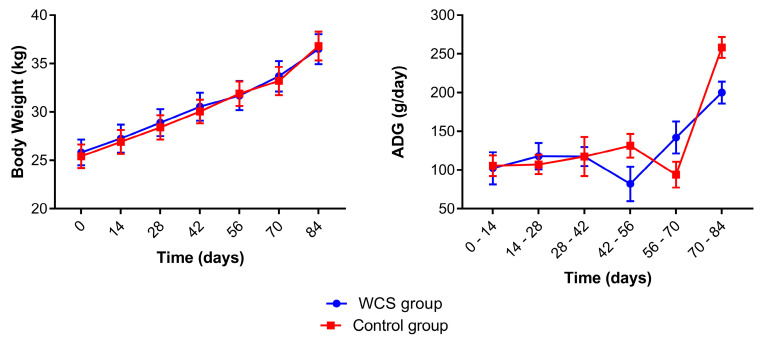
Mean (±standard error of the mean) of body weight (kg) values and average daily weight gain (ADG) of crossbred lambs naturally infected with gastrointestinal nematodes and *Eimeria* spp. receiving two different diets (whole cottonseed and control) at seven different samplings (0–84 days). There was an interval of 14 days between each sample collection.

**Figure 3 metabolites-13-00398-f003:**
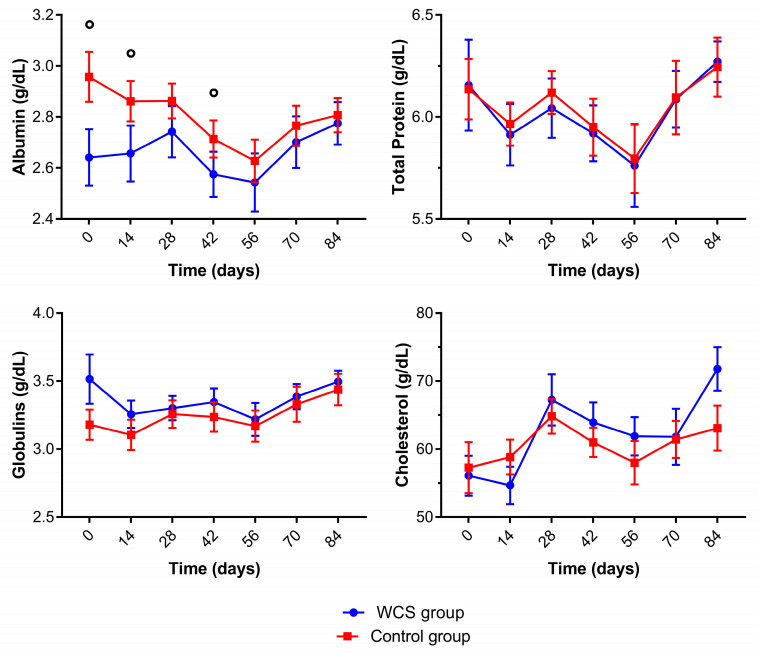
The mean (±standard error of the mean) of albumin, total protein, globulins, and cholesterol concentrations of crossbred lambs naturally infected with gastrointestinal nematodes and *Eimeria* spp. receiving two different diets (whole cottonseed and control) at seven different samplings (0–84 days). There was an interval of 14 days between each sample collection. ° Significantly difference between groups.

**Figure 4 metabolites-13-00398-f004:**
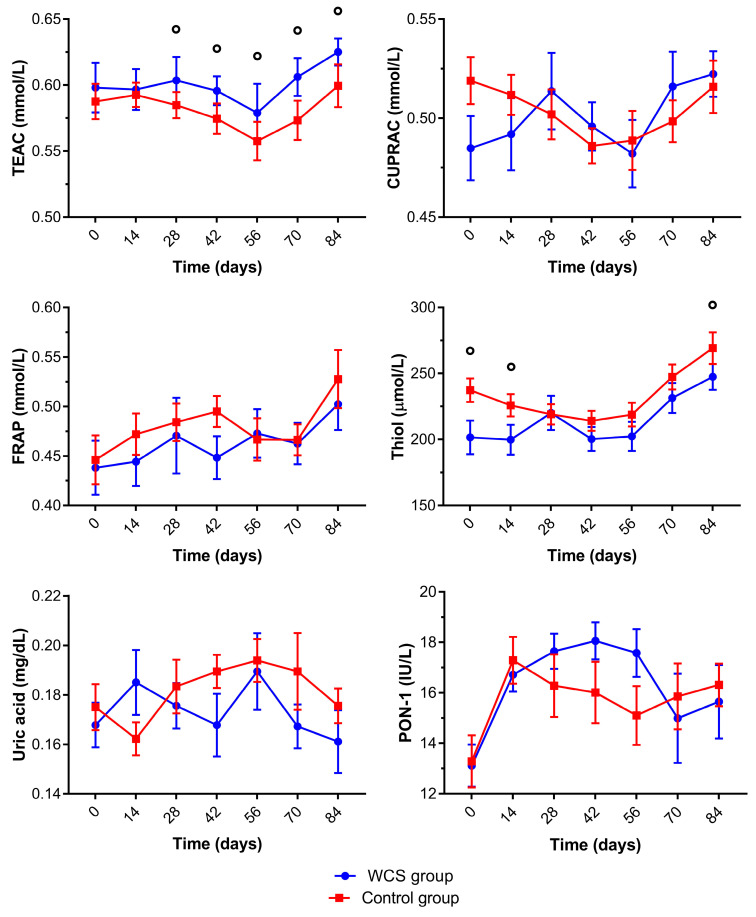
Mean (±standard error of the mean) of trolox equivalent antioxidant capacity (TEAC), cupric reducing antioxidant capacity (CUPRAC), ferric reducing ability of plasma (FRAP), thiol, uric acid, and paraoxonase-1 (PON-1) concentrations of crossbred lambs naturally infected with gastrointestinal nematodes and *Eimeria* spp. receiving two different diets (whole cottonseed and control) at seven different samplings (0–84 days). There was an interval of 14 days between each sample collection. ° Significantly difference between groups.

**Figure 5 metabolites-13-00398-f005:**
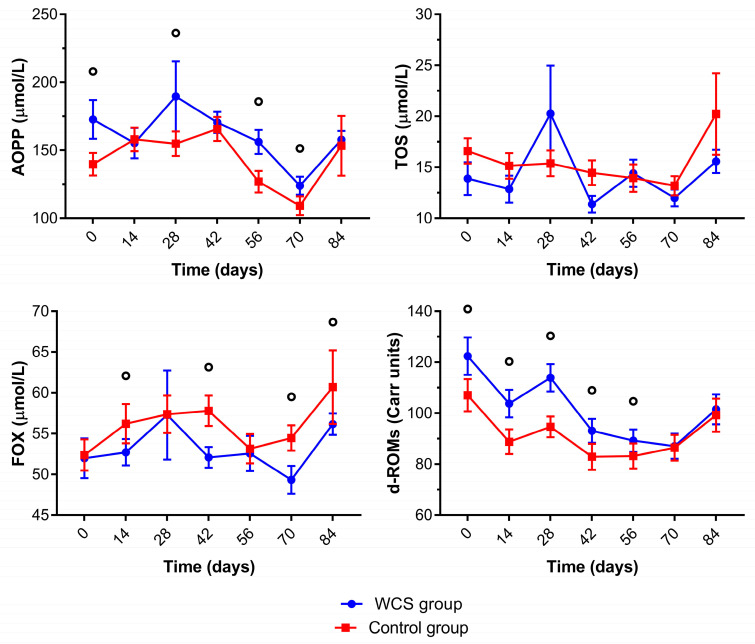
Mean (±standard error of the mean) of advanced oxidation protein products (AOPP), total oxidant status (TOS), ferric-xylenol orange (FOX), and reactive oxygen metabolites derived compounds (d-ROMs) concentrations of crossbred lambs naturally infected with gastrointestinal nematodes and *Eimeria* spp. receiving two different diets (whole cottonseed and control) at seven different samplings (0–84 days). There was an interval of 14 days between each sample collection. ° Significantly difference between groups.

## Data Availability

The data presented in this study are available in Appendix A.

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
