# Peer review of "Effect of Supplementation of Lambs with Whole Cottonseed: Impact on Serum Biomarkers and Infection by Gastrointestinal Parasites under Field Conditions"

_metabolites, 2023, doi:10.3390/metabo13030398_

Round 1

Reviewer 1 Report

The manuscript entitled “Effect of supplementation of lambs with whole cottonseed in serum biomarkers and infection by gastrointestinal nematodes and Eimeria spp. under field conditions” is interesting and it deserves publication after substantial improvement of the text and the figures. I suggest following amendments:

             The English must be improved by native speaker to make MS easily understandable and clearer.

             Title should be shortened to “Effect of supplementation of lambs with whole cottonseed on serum biomarkers and natural infection by gastrointestinal parasites”

             Abstract should be re-written: emphasize the goal of the study, omit unnecessary details from the methodology, add specific results

             Do not use "time points", use the number of days since the administration of WCS supplementation in the description of experiments, and also in the figures. Delete Fig.1 as it is unnecessary.

             Remove the information about feacal and blood sampling from sections 2.3.1. and 2.3.2 to section 2.3. (line 152), renumerate the sections 2.3.1 -2.3.3. as 2.4. - 2.6 and section Statistical analysis as 2.7.

             The heading of each subsection from the Results should begin with: The effect of WCS supplementation on….

             Delete Fig. 3. and lines 237-243. Only add information what was the average weight of lambs in the beginning and in the end of experiment.

             In all Results sections present only comparison of WCS and control groups.

             All Figures: axis x -  time (days), groups: WCS supplemented group; Control group

             Fig. 6 – make the mark for statistical significance bigger

             Line 398 (the last sentence)- omit “the combination with ICLS”  as you do not test WCS in other type of breeding

Author Response

The manuscript entitled “Effect of supplementation of lambs with whole cottonseed in serum biomarkers and infection by gastrointestinal nematodes and Eimeria spp. under field conditions” is interesting and it deserves publication after substantial improvement of the text and the figures. I suggest following amendments:

  • The English must be improved by native speaker to make MS easily understandable and clearer.

The authors would like to comment that the manuscript was sent to the American Manuscript Editors (project # 94589) for English language review and editing.

  • Title should be shortened to “Effect of supplementation of lambs with whole cottonseed on serum biomarkers and natural infection by gastrointestinal parasites”

The authors agreed to change the title and following all the reviewers comments. The title now is “Effect of supplementation of lambs with whole cottonseed: Impact on serum biomarkers and infection by gastrointestinal parasites under field conditions

  • Abstract should be re-written: emphasize the goal of the study, omit unnecessary details from the methodology, add specific results

The authors would like to mention that the abstract has been reviewed.

  • Do not use "time points", use the number of days since the administration of WCS supplementation in the description of experiments, and also in the figures. Delete Fig.1 as it is unnecessary.

The authors would like to mention that Figure 1 was deleted and that “time points” were excluded and changed to the number of days.

  • Remove the information about feacal and blood sampling from sections 2.3.1. and 2.3.2 to section 2.3. (line 152), renumerate the sections 2.3.1 -2.3.3. as 2.4. - 2.6 and section Statistical analysis as 2.7.

The authors renumerated the sections following the reviewer suggestions.

  • The heading of each subsection from the Results should begin with: The effect of WCS supplementation on….

The authors would like to comment that the results include control group as well as the WCS supplementation. Perhaps we could keep the heading of each section as they are presented to avoid misunderstanding of the results?

  • Delete Fig. 3. and lines 237-243. Only add information what was the average weight of lambs in the beginning and in the end of experiment.

The authors would like to ask to keep Figure 3 as it is relevant to observe how the mean of body weight values and average daily weight gain of both groups were affected during the experimental trial and the supplementation with WCS did not influence the final production.

  • In all Results sections present only comparison of WCS and control groups.

The authors would like to comment that the all results were reviewed to only compare WCS and control groups.

  • All Figures: axis x - time (days), groups: WCS supplemented group; Control group

The authors have changed all the figures.

  • Fig. 6 – make the mark for statistical significance bigger

The authors have improved the mark for statistical significance in all the figures.

  • Line 398 (the last sentence)- omit “the combination with ICLS” as you do not test WCS in other type of breeding

The authors would like to comment that the supplementation with WCS was part of the ICL system to test if diet improvements could be helpful in controlling gastrointestinal parasites.

The authors would like to thank the editor and reviewer for all the suggestions, comments and critical review of this paper.

Reviewer 2 Report

The manuscript entitled "Effect of supplementation of lambs with whole cottonseed in serum biomarkers and infection by gastrointestinal nematodes and Eimeria spp. under field conditions" describes the effect of ration containing cottonseed on gastrointestinal nematodes and Eimeria spp. in sheep.

I have a few comments on unclear parts.

in Materials and methods,

My main concern is that if you are evaluating the effect of your tested cotton seed ration, you should not treat with antiparasitic and anticoccidial drugs after the trial has begun.

-Are those all experimental sheep were positive with parasite?

-You should describe the intensity of parasite in all animals before the trial begins.

-You should consider all positive animals to be equally distributed in each group.

Finally (L396), you cannot conclude that "controlling the natural infection by GIN and Eimeria spp." Treatment with antiparasitic drugs during the trial period may interfere with the effect of your tested WCS.

Minor comments;

The title should be as "Effect of supplementation of lambs with whole cottonseed on serum biomarkers and infection by gastrointestinal nematodes and Eimeria spp. under field conditions".

Author Response

Dear Reviewer,

Thank you for your comments. The manuscript was reviewed by the American Manuscript Editors. Please see the authors responses below:

1. Materials and methods

My main concern is that if you are evaluating the effect of your tested cotton seed ration, you should not treat with antiparasitic and anticoccidial drugs after the trial has begun.

-Are those all experimental sheep were positive with parasite?

-You should describe the intensity of parasite in all animals before the trial begins.

-You should consider all positive animals to be equally distributed in each group.

All sheep were evaluated for parasite infection before the trial as day 0 (Figure 1) shows the results of parasite intensity. Groups were balanced as closely as possible by body weight as the aim of the study was to evaluate the diet supplementation with whole cotton seed. During the trial the animals were only treated with an anticoccidial due to the unexpected Eimeria spp. infection (please see comment below). Animals were not treated with antiparasitic drugs for nematode parasites during the trial as it was expected that the ICL system would reduce their parasitic burden (Almeida et al., 2018; Schmidt et al., 2021)

2. Finally (L396), you cannot conclude that "controlling the natural infection by GIN and Eimeria spp." Treatment with antiparasitic drugs during the trial period may interfere with the effect of your tested WCS.

The Eimeria spp. peak of OPG counts at 42 days for both treatments was not expected. We decided to treat all animals and to perform bedding changes as well as cleaning of pens and troughs for helping the control the infection and to reduce the environmental contamination following the practices of animal health and welfare. 

Line 396 (now line 358):  We concluded that the WCS inclusion suggested a benefit to controlling the natural infection by GIN and Eimeria spp in this trial.

3. Minor comments

The title should be as "Effect of supplementation of lambs with whole cottonseed on serum biomarkers and infection by gastrointestinal nematodes and Eimeria spp. under field conditions".

The title has been changed to: "Effect of supplementation of lambs with whole cottonseed: Impact on serum biomarkers and infection by gastrointestinal parasites under field conditions" following the other reviewers suggestions.

The authors would like to thank the editor and reviewer for all the suggestions, comments and critical review of this paper. Please find attached the reviewed PDF file.

Reviewer 3 Report

The authors did a great job in presenting this interestingstudy. As there are many problems nowadays in controlling parasitic diseases, especially GIN and protozoans due to bad management, unsuitable treatment or more commonly lack of efficiency of some molecules, this study brings new information about controlling these infections by using WCS in the diet of lambs.

The authors proved that WCS could help reduce oxidative stress and control the endoparasites in lambs.

Overall, the paper is well-written, easy to follow, and understandable.

I find it suitable for publication in its present form.

Author Response

Dear Reviewer,

Thank you for your comments. Please find attached the reviewed PDF file.

The authors would like to thank the editor and reviewer for all the suggestions, comments and critical review of this paper.

Reviewer 4 Report

Dear authors:

The research investigation was well-conducted; all controls and time points evaluated look tremendous and support the findings.

Although, the groups treated with the dietary supplement and the control group in all parameters evaluated are similar. Indeed, some parameters show that combining whole cottonseed (WCS) in an integrated crop-livestock system (ICLS) can be very beneficial in controlling the parasitic infection.

There is only a minimum of editing errors that I am sure the authors could edit:

Lane 108. Remove the dot before the subtitle

Scientific names should be italicized, for example:

Lane 116. Clostridium

Lanes 117, 127, 384, 405, 412. Eimeria

Author Response

Dear Reviewer,

Thank you for your comments. The manuscript has been reviewed by American Manuscript Editors for English language and style revisions.

Lane 108. Remove the dot before the subtitle - we removed the dot (now line 105)

Scientific names should be italicized, for example:

Lane 116. Clostridium; Lanes 117, 127, 384, 405, 412. Eimeria - all scientific names have been reviewed.

Please see the reviewed PDF file attached.

The authors would like to thank the editor and reviewer for all the suggestions, comments and critical review of this paper.

Round 2

Reviewer 2 Report

During trial period, the tested animals were treated with an anticoccidial due to the unexpected Eimeria spp. infection, therefore, the conclusion should not include about "controlling Eimeria". Only you can conclude about the effect on nematode.

In line 34, 357  please delete "Eimeria".

In line 113, 148 199; did you check all those parasite eggs are nematode? Small ruminants often infested with other cestode such as Monieza spp. If the sheep were infested with nematode and cestode, you should use "helminth". Please check it.

Author Response

Dear Reviewer, 

Thank you for your comments. 

In line 34, 357  please delete "Eimeria". The authors have changed line 34 to: "The inclusion of WCS suggested the benefit in controlling infection as well as inducing an increase in antioxidants and a decrease in oxidants in lambs naturally infected by gastrointestinal parasites" and line 357 to:"The WCS inclusion in this trial suggested a benefit to controlling the natural infection by GIN  as well as inducing a decrease in oxidants such as TOS and FOX and an increase in the antioxidant TEAC in lambs".

In line 113, 148 199; did you check all those parasite eggs are nematode? Small ruminants often infested with other cestode such as Monieza spp. If the sheep were infested with nematode and cestode, you should use "helminth". Please check it. The authors would like to mention that all the results were double checked and the eggs were from gastrointestinal nematode parasites and  Eimeria spp. oocysts . No Moniezia spp. eggs were found.
